# Membrane-associated σ factors disrupt rRNA operon clustering in *Escherichia coli*

**Khang Ho**\*, **Rasika M. Harshey**(ORCID)\*

Department of Molecular Biosciences and LaMontagne Center for Infectious Diseases, The University of Texas at Austin, Austin, Texas, United States of America

\* khang.ho@utexas.edu (KH); rasika@austin.utexas.edu (RMH)

## Abstract

Chromosomal organization in *Escherichia coli* as examined by Hi-C methodology indicates that long-range interactions are sparse. Yet, spatial co-localization or "clustering" of 6/7 ribosomal RNA (*rrn*) operons distributed over half the 4.6 Mbp genome has been captured by two other methodologies—fluorescence microscopy and Mu transposition. Our current understanding of the mechanism of clustering is limited to mapping essential *cis* elements. To identify *trans* elements, we resorted to perturbing the system by chemical and physical means and observed that heat shock disrupts clustering. Levels of σ$^H$ are known to rise as a cellular response to the shock. We show that elevated expression of σ$^H$ alone is sufficient to disrupt clustering, independent of heat stress. The anti-clustering activity of σ$^H$ does not depend on its transcriptional activity but requires core-RNAP interaction and DNA-binding activities. This activity of σ$^H$ is suppressed by ectopic expression of σ$^D$ suggesting a competition for core-RNAP. A query of the other five known σ factors of *E. coli* found that elevated expression of FecI, the ECF σ factor that controls iron citrate transport, also perturbs clustering and is also suppressed by σ$^D$. We discuss a possible scenario for how these membrane-associated σ factors participate in clustering of distant *rrn* loci.

## Introduction

Bacteria inhabit every available niche on earth, where they are subject to a range of environmental conditions to which they must acclimatize [1]. When conditions are favorable, bacteria quickly synthesize proteins required for uptake and biosynthesis of cellular building blocks that enable virtually every aspect of cell growth [2]. The bulk of cellular transcription during this phase is dedicated to ribosomal RNA (*rrn*) operons that synthesize ribosomes. Not only are *rrn* operons highly transcribed, but most bacteria also possess multiple such operons. The multiplicity of *rrn* operons has been correlated with elevated growth rate, genome integrity, and acquisition of more diverse biosynthetic pathways [3,4], suggesting that there is an evolutionary advantage to maintaining multiple copies of these operons.

**Data availability statement:** All relevant data are within the paper and its Supporting information files.

**Funding:** This work was primarily funded by National Institutes of Health grant GM118085 to RMH. Partial support was provided by Robert Welch Foundation (F-2190) to RMH. The funders had no role in study design, data collection and analysis, decision to publish, or preparation of the manuscript.

**Competing interests:** The authors have declared that no competing interests exist.

**Abbreviations :** ECF, extracytoplasmic function; IPTG, Isopropyl-β-ᴅ-Galactoside; ONPG, O-nitrophenyl-β-D-Galactoside; PNK, polynucleotide kinase; PBS, phosphate-buffered saline; *rrn*, ribosomal RNA.

There are seven *rrn* operons in *Escherichia coli* (*rrnA-G*), distributed on both arms (replicores) of the bi-directionally replicating chromosome, and residing in the upper half of each replicore [5] (S1 Fig). FROS (Fluorescent Reporter Operator Sites) experiments using pairwise *parS*-ParB interactions showed that 6/7 *rrn* loci spatially co-localize or cluster, reminiscent of the eukaryotic nucleolus [6]. This cluster was not detected by the more widely used Hi-C methodology, which employs formaldehyde to crosslink chromosomal interactions bridged by proteins [7]. Failure to detect the *rrn* cluster by this method could be due to disruption of the cluster by formaldehyde, or to a distance unfavorable for crosslinking. An alternative crosslinking method that exploits the natural mechanism of phage Mu transposition to link distant DNA sites indeed detected the *rrn* cluster [8]. The Mu method is not widely used as yet, because of the limited host-range of Mu, but has been additionally validated by corroborating the existence of a distinct Ter region on the *E. coli* genome [9] as demonstrated using other techniques [7,10].

The FROS study delineated *cis*-acting elements required for *rrn* clustering by systematically deleting regulatory regions upstream of *rrnD* and monitoring its co-localization with *rrnG*, both located on the same arm of the chromosome but separated by 700 Kbp (S1 Fig) [6]. The study found that clustering required P1, the stronger of the two promoters driving transcription of *rrnD*, as well as an upstream binding site for the NAP (Nucleoid Associated Protein) Fis, but that neither Fis nor other NAPs known to regulate *rrn* transcription were required, suggesting that transcription of the *rrn* locus was not responsible for clustering. Consistent with this notion, mutation of the conserved −10 region of P1 failed to disrupt clustering, indicating that the formation of an open complex was also dispensable. Taken together, these results indicated that multiple transcribing RNA polymerases expected at these highly transcribed loci are likely not the cause of clustering. The Mu method showed that the NAP HUα and the condensin Muk-BEF influenced cluster formation, likely by affecting chromosome compaction in general [8].

To gain more insight into the phenomenon of *rrn* clustering we attempted to perturb the system by subjecting cells to amino acid starvation, as well as to heat, cold and ethanol shock. Of these, heat shock completely disrupted clustering. The heat shock response, part of the more general unfolded protein response [11], is designed to maintain heat-denatured proteins in a properly folded state, a key player in this response being σ$^H$ [12]. σ$^H$ (RpoH) regulates a large number of genes, notably those encoding protein chaperones such as GroEL/GroES [13,14]. We show that it is the rise in σ$^H$ levels and not heat stress per se, that disrupts clustering, and we corroborate this observation using the Mu method. The observed σ$^H$-promoted de-clustering could be counteracted by simultaneous expression of σ$^D$ (or σ$^{70}$). A similar but weaker effect on de-clustering was exhibited by FecI, which was also rescued by σ$^D$. Both σ$^H$ and FecI are associated with the inner membrane. Based on these findings, we propose a model for how clustering of *rrn* operons occurs at the membrane and might be driven by the ability of sigma factor(s) to assemble RNAP onto cognate promoters.

## Results

### Heat shock disrupts *rrnA-rrnD* clustering

Gaal and colleagues used FROS to examine pair-wise combinations of 15 of the possible 21 pairs of *rrn* operons, using distinct *parS*-ParB partners derived from phage P1 and plasmid pMT1 of *Yersinia pestis* [6,15]. In this study, we employed the same approach to determine *trans*-acting factors involved in clustering (Fig 1A). For our analysis we chose *rrnA* and *rrnD*, located on two different replicores (see S1 and S2 Figs for their relative location and for loci images, respectively), directing ParB-GFP 361 bp upstream of *rrnA* and ParB-CFP 282 bp upstream of *rrnD* by inserting their respective *parS* sites at these locations; the median *rrnA-D* distance was estimated to be 134 nm. These cells were then subjected to various well-studied chemical and physical stresses (Fig 1B). Serine hydroxamate (SHX) is a serine analog that inhibits tRNA^Ser aminoacylation, mimicking amino acid starvation and inducing a stringent response with concomitant synthesis of (p)ppGpp [16,17]. Application of this stress failed to produce a notable change (>2-fold) in the median *rrnA-D* distance (214 nm, compared to a significance cutoff at 230 nm; see figure legend for assignment of significance). Cold shock, which elicits changes in membrane fluidity, protein and nucleic acid folding and ribosome assembly [18], also did not significantly perturb clustering (216 nm). We note that since the resolution of our methods does not enable us to meaningfully distinguish distances below approximately 250 nm, distances below the cutoff are considered the same. Ethanol damages cell wall and membrane integrity, inducing an unfolded protein response in addition [11,19]. This stress increased the *rrnA-rrnD* distance to 424 nm. The unfolded protein response can also be produced by heat stress, which produced the most

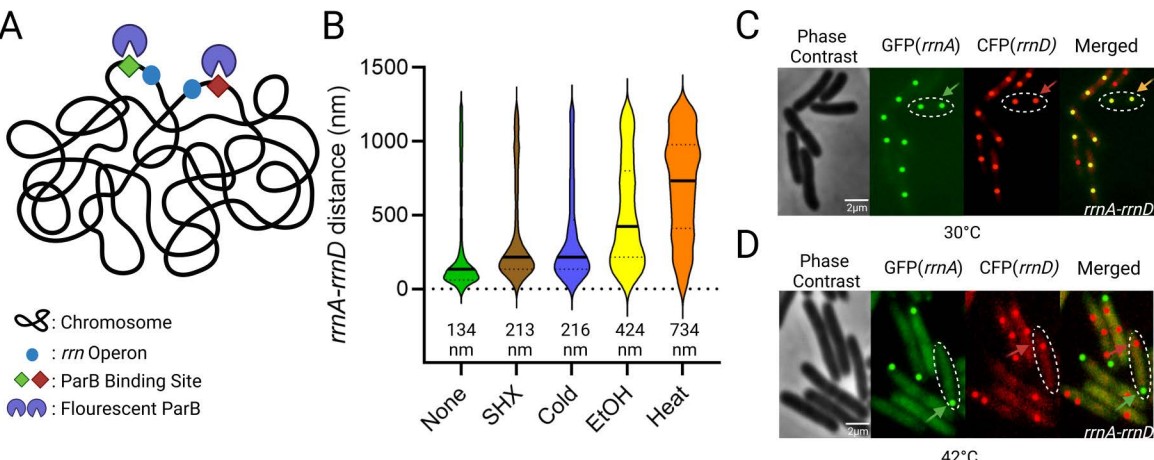

**Fig 1. Distance between *rrnA* and *rrnD* operons under various stress conditions. (A)** Scheme for *rrn* operon tagging. Two different par sites (*parSpMT* and *parSP1*) were placed upstream of *rrnA* and *rrnD*, respectively, in the parent strain MG1655. These sites were visualized by co-expression of their cognate fluorescent ParB proteins (pMT1 GFP-ParB and P1 CFP-ParB). **(B)** Violin plots of the distance between *rrnA-rrnD* under indicated stress conditions (see Methods for details and S1 Table for strains and plasmids). The median distance (nm) between labeled loci is given below each graph. Between 400 and 900 foci are shown for each sample. The solid line indicates the median distance, and the top and bottom dashed lines indicate the third and first quartile, respectively. We note that due to the large number of foci observed, a small change in the median distance is considered statistically significant (*p* < 0.001) under Mann–Whitney test. The data shown are for one of three biological replicates. We arbitrarily considered a 2.5-fold change of median distance to be significant. We also note that since we have values that are 0, the border tries to encompass these data points, giving a false impression of negative distances. Data used for plotting this, and all other graphs can be found in the Supplemental "S1 Data" file under appropriate figure headings. **(C)** A representative image of *rrnA-rrnD* foci without added stress; many such images were used to generate data in **(B)**. GFP and CFP were false-colored and enhanced for better visualization. One cell is outlined, with an arrow pointing to a merged GFP/CFP focus. The foci are edited in post-processing as a perfect circle to provide better contrast and visualization. Most cells appear to contain two copies of the *rrn* operons, indicating that this region of the chromosome is replicated. **(D)** Representative image of *rrnA-rrnD* de-clustering with heat stress. Panel labels as in C. Colored arrows indicate focus from either GFP-field (green) or CFP (red). We note that the number of *rrnA*-GFP foci is reduced to 1 focus per cell. See Methods for the code used to analyze fluorescent foci in this, and all similar figures. Created in BioRender. Ho, K. (2025) https://BioRender.com/p78s829.

pronounced shift of the median distance relative to control (734 nm) (Fig 1B). Representative images from these experiments are shown in Fig 1C and 1D. We note that heat stress appears to localize *rrnA*, but not *rrnD*, to the pole, while also reducing the number of *rrnA* loci to 1 (Fig 1D, green arrow). Since ethanol also induces the unfolded protein response, we infer that it is the unfolded protein response that promotes de-clustering of *rrnA* and *rrnD*.

Since our laboratory has shown that long-range contacts occur less frequently in an *hupA* (HUα) mutant [8], we also examined a noncoding (nc) RNA known to interact with HU [20]. A single deletion of nc5 showed no difference in the median distance between *rrnA* and *rrnD* (S3 Fig). RpoZ (an RNAP subunit) and NusB (a component of the anti-termination complex that interacts with RNAP), reported to contribute to phase-separation of *E. coli* RNAP [21], were also examined. Deletion of either *rpoZ* or *nusB* elicited a small increase (i.e., above our 2-fold cutoff of 230 nm) in the *rrnA-D* distance (254 nm and 304 nm, respectively) (S3 Fig), but not as drastic as that of heat stress.

In summary, of all the variables tested, heat stress caused the most significant de-clustering of the *rrnA-D* pair, followed by ethanol stress. These two stresses share the common outcome of producing an unfolded protein response.

## Deregulation of GroEL/S disrupts *rrnA-rrnD* clustering

Since heat shock disrupted the *rrnA-D* pair, we hypothesized that some protein factor(s) bridging the two loci was displaced as a result. To identify bound proteins, we directed dCas9 (a variant of Cas9 capable of binding but not cleavage [22]), to upstream regions of all seven *rrn* loci, similar to the location of *parS* sites (see S2 Table), using the appropriate gRNAs. dCas9 was fused to 3X-FLAG, so anti-FLAG antibodies were used in pulldown experiments (S4A Fig). The efficiency of sgRNA targeting was determined by assessing lethality when Cas9 was provided instead of dCas9 (S5 Fig). Overall, in a host expressing Cas9, sgRNAs directed to all *rrn* operons caused a reduction in viability relative to a no-PAM control (S5 Fig). The differences in gRNA efficiency may influence the efficiency of dCas9 pulldowns, but since Cas9 targeting was productive, we concluded that the pulldowns would reflect at least the binding of dCas9 to the rDNA.

Mass spectrometry of the proteins identified in pulldowns with the seven *rrn* samples showed a range of significantly enriched proteins across different sgRNAs (*z*-score > 2.5, computed from three biological replicates for each sgRNA) using the methods described in [23]. For sgRNA targeting *rrnA*, *rrnB*, *rrnC*, *rrnD*, *rrnE*, *rrnG*, and *rrnH*, we found 54, 132, 135, 105, 66, 33 and 107 significantly enriched proteins, respectively (S4B and S6 Figs). Common proteins identified across multiple samples are summarized in S7 Fig. The detailed spectral counts of identified proteins for each sgRNA can be found in S3–S9 Tables. No common protein was significantly enriched across all samples, but GroEL was enriched in 4/7 sgRNAs (*rrnC*, *rrnA*, *rrnB*, and *rrnD*), suggesting that GroEL could be a possible *trans*-acting factor in clustering. To control background protein abundance, we normalized *rrn* targeting samples to a gRNA targeting the non-*rrn* gene *lacZ* (with PAM) (see Methods). A potential pitfall of this approach is that we could not verify that clustering was preserved after formaldehyde crosslinking, due to loss of resolution of ParB foci after fixation, so it is possible that our data may not have sufficiently captured all the major players in clustering.

GroEL, in complex with GroES, acts as a chaperone for protein folding [24]. To investigate whether GroEL mediated the *rrn* clustering, we generated a chromosomal *groEL* deletion in the *parS*-tagged *rrnA-D* strain (S4C Fig). Since *groEL* is essential [25], we provided it in *trans* from a plasmid under control of pAraBAD and verified induction (with arabinose) and repression (with glucose) by observing corresponding increases and decreases in colony sizes, respectively (S8 Fig). Examination of the degree of *rrnA-rrnD* clustering under these two conditions showed that clustering was disrupted with arabinose addition (S4 Fig, third plot, red, from the left), even though GroEL levels were sufficient for growth as judged by colony size (S8 Fig); the median distance between *rrnA-D* increased to 494 nm. A WT 'control' carrying both *groEL*/*groES* under pAraBAD control increased *rrnA-D* distance to 216 nm (S4C Fig, second plot, yellow, from left); this slight increase is not significant based on our 2-fold cut-off, but the data nonetheless suggest that ectopic expression *of groES*/*EL* affects clustering. Taken together, these results suggest that any perturbation of normal GroES/EL levels destabilizes clustering. Since these two genes are co-transcribed [26], we wondered if the imbalance in their relative levels was responsible for

cluster disruption. We therefore placed the entire *groES*/*EL* operon on the plasmid vector in the Δ*groEL* strain, but that did not restore clustering either (S4C Fig, rightmost plot).

To test if expression of *groES*/*groEL* from their native σ^H-promoter (pGroE) would change the results, we compared *rrnA-D* distance when expression of this operon was controlled by pGroE versus pAra in both WT and Δ*groEL* strains (S4D Fig). In WT, pGroE had no discernable effect. In the Δ*groEL* strain, however, the *rrnA-D* distance shifted closer to normal with the pGroE plasmid as indicated by the 25% quartile of 242 nm for pGroE versus 356 nm for pAra (S4D Fig, compare rightmost two plots), suggesting that transcription from the native promoter was better at restoring clustering.

σ^H, also known as σ$^{32}$ or RpoH, is the major heat shock sigma factor that transcribes the *groEL*/*S* operon exclusively [26]. Its levels are kept low through multiple mechanisms, including sequestration at the inner membrane, and direct interaction of GroEL/ES with σ^H [27,28]. Could our inability to restore *rrnA-rrnD* clustering in the Δ*groEL*/*ES* background be attributable to perturbation of σ^H levels? This was tested next.

## σ^H disrupts *rrnA-rrnD* independent of its transcriptional activity

If perturbation of σ^H was the cause of *rrnA-D* de-clustering, we hypothesized that ectopic expression of σ^H alone should give similar results. We therefore placed *rpoH* under an inducible Tet promoter (pTc). Consistent with our expectations, *rpoH* induction increased the median distance between *rrnA-D* to 566 nm (Fig 2A, third plot from left). However, the same result was seen even in the absence of *rpoH* induction (Fig 2A, second plot from left). Unlike heat stress, *rpoH* did not affect the number of *rrnA* foci (S9 Fig, compare to Fig 1D). To verify that *rpoH* was being expressed from the plasmid, we constructed a *lacZ* reporter driven from p*htpG*, a weak σ^H-responsive promoter [29], and assessed *lacZ* expression upon induction of *rpoH* from pTc (Fig 2B). A small but significant increase in β-galactosidase activity was observed after an hour of induction, the same time frame employed for microscopy (Fig 2A), indicating that *rpoH* was expressed from the Tet promoter. We conclude that even a small increase in σ^H levels promotes de-clustering of *rrnA-rrnD*.

To ascertain that the data in Fig 2A are not merely due to perturbation of cellular morphology and nucleoid structure, we measured both, under both heat stress and *rpoH* induction. Under the heat stress, cell lengths showed a small but statistically significant increase: 2.9 μm on average compared to 2.8 μm for the control (S10A Fig). Under *rpoH* overexpression, cell lengths decreased on average to 2.5 μm compared to 2.8 μm for the control. To determine the effect of these perturbations on the nucleoid, we performed DAPI staining under these conditions. Curiously, we found that the nucleoid is more condensed, not decondensed, under both conditions (S10B Fig). These results indicate that the de-clustering effect of heat stress or *rpoH* expression is not a consequence of changes in nucleoid compaction or cellular morphology. As another control to ascertain that the change in *rrnA-D* distance was specific to these loci and not due to a general increase in distance between any two loci, we tagged *lacZ*-pMT*parS* and *leuC*-P1*parS* (200 Kbp apart; S10C Fig, left) and measured their distance −/+ *rpoH* expression (S10C Fig, right). We observed no change in their distance between the two conditions.

*E. coli* possesses seven experimentally confirmed sigma factors, of which the vegetative σ^D (RpoD or σ$^{70}$) is the most abundant [30]. Since the amount of RNAP is thought to be relatively constant across most growth conditions, competition among the sigma factors for this core RNAP has been proposed to be the mechanism for gene regulation [30,31]. To test if competition between σ^H and σ^D for binding RNAP could be responsible for destabilizing clustering, we inserted *rpoD* downstream of TetR on the same plasmid, which is driven by the constitutive promoter divergent from pTc (pTetR). Expression of *rpoD* decreased the median distance between *rrnA-D* to 213 nm compared to 566 nm with the *rpoH* vector alone (Fig 2A, compare fourth plot to second plot from the left), supporting the notion that competition between σ^D and σ^H for core RNAP likely suppressed the de-clustering activity of σ^H.

To test this notion further, we examined the following previously characterized mutants of *rpoH*, reported to have decreased or increased transcription activities: L245P is defective for interaction with RNAP [32], E265A is defective for

promoter binding at the −35 region [32], and I54N has increased stability [28,29]. Our expectation was that the L245P and E265A would lose their de-clustering activity, while I54N mutant would not. This expectation was borne out for the L245P and E265A variants (303 nm and 216 nm, respectively), but not for I54N (284 nm) all three variants were defective in destabilizing the *rrn* cluster, compared to WT RpoH (566 nm) (Fig 2C). We note that in contrast to cells expressing the first two variants, those expressing I54N are elongated (S11 Fig). I54N has been shown to exhibit elevated transcriptional activity due to its inability to localize to the membrane like WT σ^H [27,29], which likely perturbs normal cell physiology. The expected transcriptional activity of all the σ^H mutants was confirmed using the *lacZ* reporter as before (Fig 2D). We conclude that σ^H -mediated de-clustering is independent of its transcriptional activity.

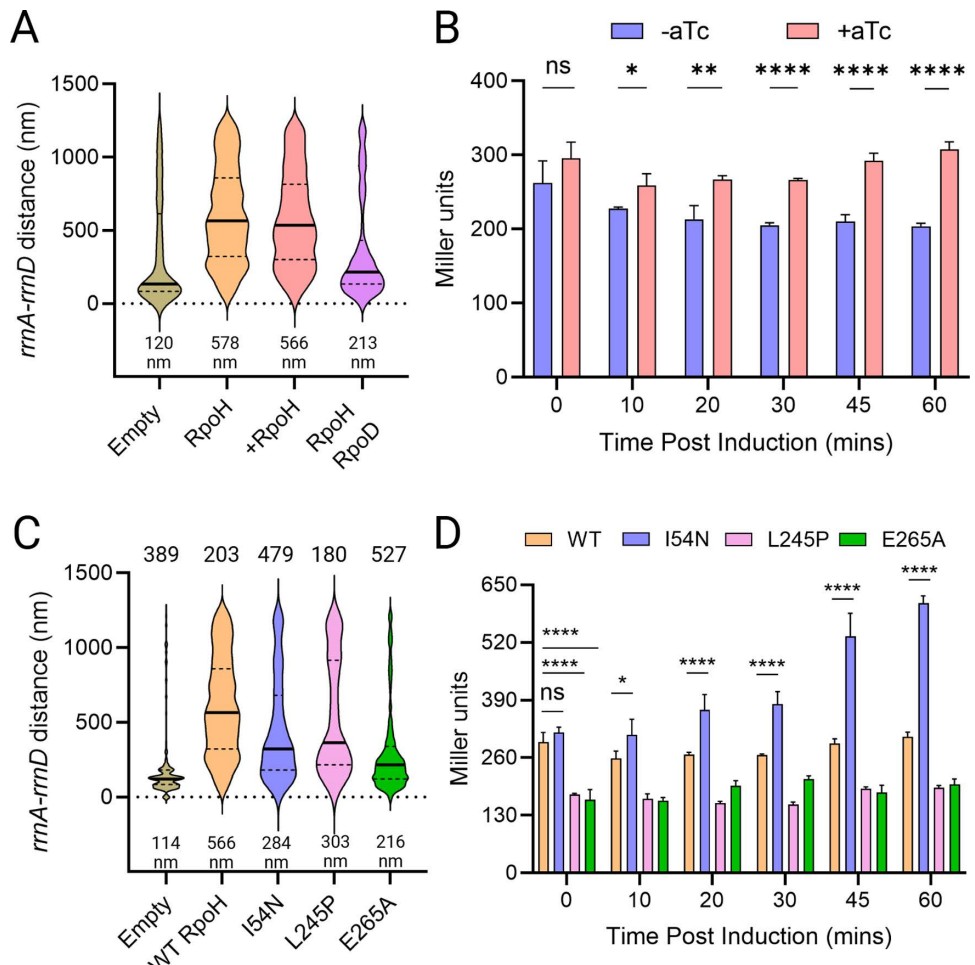

**Fig 2. σ^H promotes de-clustering of *rrnA-rrnD* independent of its transcriptional activity.** **(A)** Distance between *rrnA-D* with ectopic expression of *rpoH* from pTc (+). Induction was carried out for one hour prior to microscopy. In the fourth plot from left, *rpoD* is cloned downstream of TetR, from a constitutively expressed promoter, divergent from pTc. All other descriptions as in Fig 1. **(B)** Transcriptional activity of ectopically expressed *rpoH*. Miller assay was carried out as described in Methods. Student t's test was performed pairwise to determine statistical significance (two-tailed, ns: not statistically significant, *p < 0.05, **p < 0.01, ****p < 0.0001). **(C)** Distance between *rrnA-D* with ectopic expression of indicated *rpoH* mutants, compared to empty vector control. **(D)** Transcriptional activity of ectopically expressed *rpoH* mutants, as described in B. Data used for plotting Fig 2 graphs can be found in the Supplemental "S1 Data" file under appropriate figure headings. Created in BioRender. Ho, K. (2025) https://BioRender.com/k54q156.

## σ^H does not disrupt all *rrn* pairs

Our experiments thus far queried the co-localization of the *rrnA-D* pair as representative of all the *rrn* loci found in the cluster [6]. To test if heat shock and σ^H were equivalent in their disruptive ability, we decided to query other pairs, including every co-localizing *rrn* operon at least once, with *rrnC* as the non-clustering control. Heat shock disrupted clustering of all non-C *rrn* operons tested, the median distance increasing 1.5–2.5 fold (from 80–120 nm to 450–700 nm) (Fig 3A; summary of the results is diagrammed in Fig 3B). *rrnC* is reported to not be part of the cluster [6,8], yet its distance from *rrnG* remained curiously unperturbed by heat, staying at ~400 nm median, suggesting perhaps a specificity to clustering that is immune to the global effect of heat stress.

The effect of RpoH on the *rrn* pairs tested was non-uniform (Fig 3C). Specifically, we observed that two pairs that shared *rrnH* appeared to be resistant to disruption by σ^H (Fig 3C). Thus, contacts between all the *rrn* loci in the cluster are not uniform, suggesting that there is likely a sub-organization within this structure.

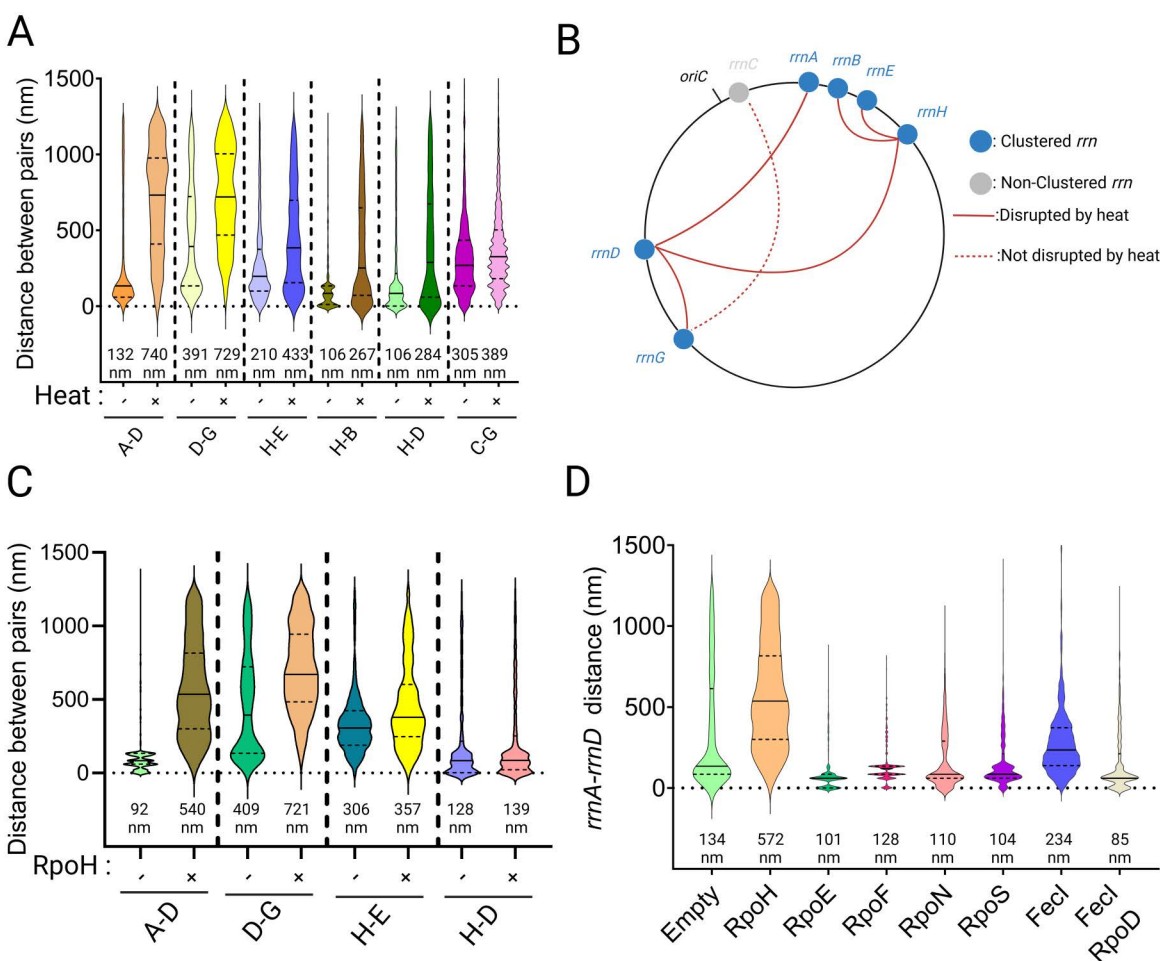

**Fig 3. Disruption of *rrn* clustering by heat stress and by FecI. (A)** Effect of heat stress on clustering of indicated *rrn* pairs. Other descriptions as in Fig 1. **(B)** Schematic summary of data in A. Each of the 7 *rrn* operons was visited at least once. **(C)** Response to *rpoH* induction of tested *rrn* pairs. +/− symbols are self-explanatory. **(D)** Effect of induced ectopic expression of five σ factors on *rrnA-D* distance, all placed under control of pTc with the same RBS. Data used for plotting Fig 3 graphs can be found in the Supplemental "S1 Data" file under appropriate figure headings. Created in BioRender. Ho, K. (2025) https://BioRender.com/g63t231.

To determine if σ^H played a unique role in affecting cluster stability, we ectopically expressed the other five sigma factors: RpoE (σ²⁴), RpoF (σ²⁸), RpoN (σ⁵⁴), RpoS (σ³⁸) and FecI (σ¹⁹), under the same promoter as the *rpoH*-expressing vector, maintaining the same RBS (ribosome binding site). Of these, FecI produced a smaller (compared to RpoH) desta-bilizing effect (134 nm versus 234 nm; Fig 3D). To test if σ^D would restore clustering to the FecI-expressing strain like it did when co-expressed with σ^H (Fig. 2A), we co-expressed *rpoD* with *fecI*; clustering was completely restored to WT levels (Fig 3D, compare last two plots). We note that the shared properties of RpoH and FecI are that they are both normally sequestered at the inner membrane, suggesting perhaps that *rrn* clustering may have a membrane component.

### σ^H disrupts the *rrn* cluster as tracked by the Mu method

Phage Mu transposition requires direct contact between Mu and its transposition target and displays virtually no sequence specificity in its target choice [33]. Higher or lower frequencies of transposition are therefore interpreted to reflect higher or lower rates of physical contact between the interacting chromosomal regions, analogous to the contact frequencies inferred from normalized Hi-C data [34]. Contacts made by Mu when located next to *rrnD* showed significantly positive interaction between *rrnD* and chromosomal regions containing *rrnA*, *B*, *E*, *G*, and *H* (but not *rrnC*), suggesting physical proximity of these regions, as also seen by FROS [6].

Since we did not examine the effect of σ^H on all possible combinations of *rrn* loci with FROS (Fig 3C), we used the Mu method to examine transposition patterns of Mu located near *rrnD* in the presence of ectopically expressed RpoH. The pattern of Mu transposition from the *rrnD*-proximal locus in WT cells was consistent with the earlier report, where Mu could access every region of the chromosome irrespective of its starting location (Fig 4A). In the presence of the vector encoding RpoH, even without induction of RpoH expression, the transposition landscape was altered, with more Mu insertions now occurring proximal to the starting Mu in Bin 73 (Fig 4B). This increase in local versus distal contacts is exacerbated with *rpoH* induction (Fig 4C). Transposition frequencies across different conditions are summarized in Fig 4D. We observed that with expression of *rpoH*, Mu predominantly inserts in its starting bin (yellow bar, indicating highest transposition frequency, seen only in the RpoH and +RpoH data), with insertions to distant bins decreased (darker blues in +RpoH and RpoH columns compared to No Vector) (Fig 4D). We conclude that *rpoH* overexpression disrupts long-range contacts.

We next specifically examined the bins containing *rrn* operons to assess the effect of RpoH on Mu transposition. We observed that contact between *rrnD* and a majority of the *rrn* loci decreased (Fig 4E, 4F, 4H, and 4J), with the notable exception of those with *rrnC* and *rrnH*, which remained unchanged (Fig 4G and 4I). The *rrnC* result is consistent with ear-lier reports [6,8], and the *rrnH* result is consistent with data in Fig 3C, where RpoH promoted de-clustering of all *rrn* exam-ined, with the exception of *rrnH*. Overall, the Mu transposition data support the hypothesis that σ^H promotes de-clustering of *rrn* operons.

## Discussion

This study demonstrates that the *E. coli* "nucleolus" where 6/7 *rrn* operons are reported to co-localize, can be perturbed by the cellular heat shock response as demonstrated by both FROS and Mu transposition. This perturbation is not due to heat stress per se, but rather due to elevation of RpoH or σ^H levels known to occur during the response. We discuss below what this and other data may suggest about the mechanism of co-localization of the *rrn* operons.

### Ability to disrupt its organization validates the existence of the *E. coli* 'nucleolus'

In eukaryotic cells, the nucleolus is the primary site of ribosome biogenesis [35]. It is a large structure, where hundreds to thousands (depending on the organism) [36] of rRNA operons localize, a device thought to have evolved to maximize translation efficiency [35]. By contrast, *E. coli* has only seven operons encoding ribosomal DNA (*rrn* operons) [37] (S1 Fig), 6/7 of which were observed to come together in some unknown fashion as reported by two completely different

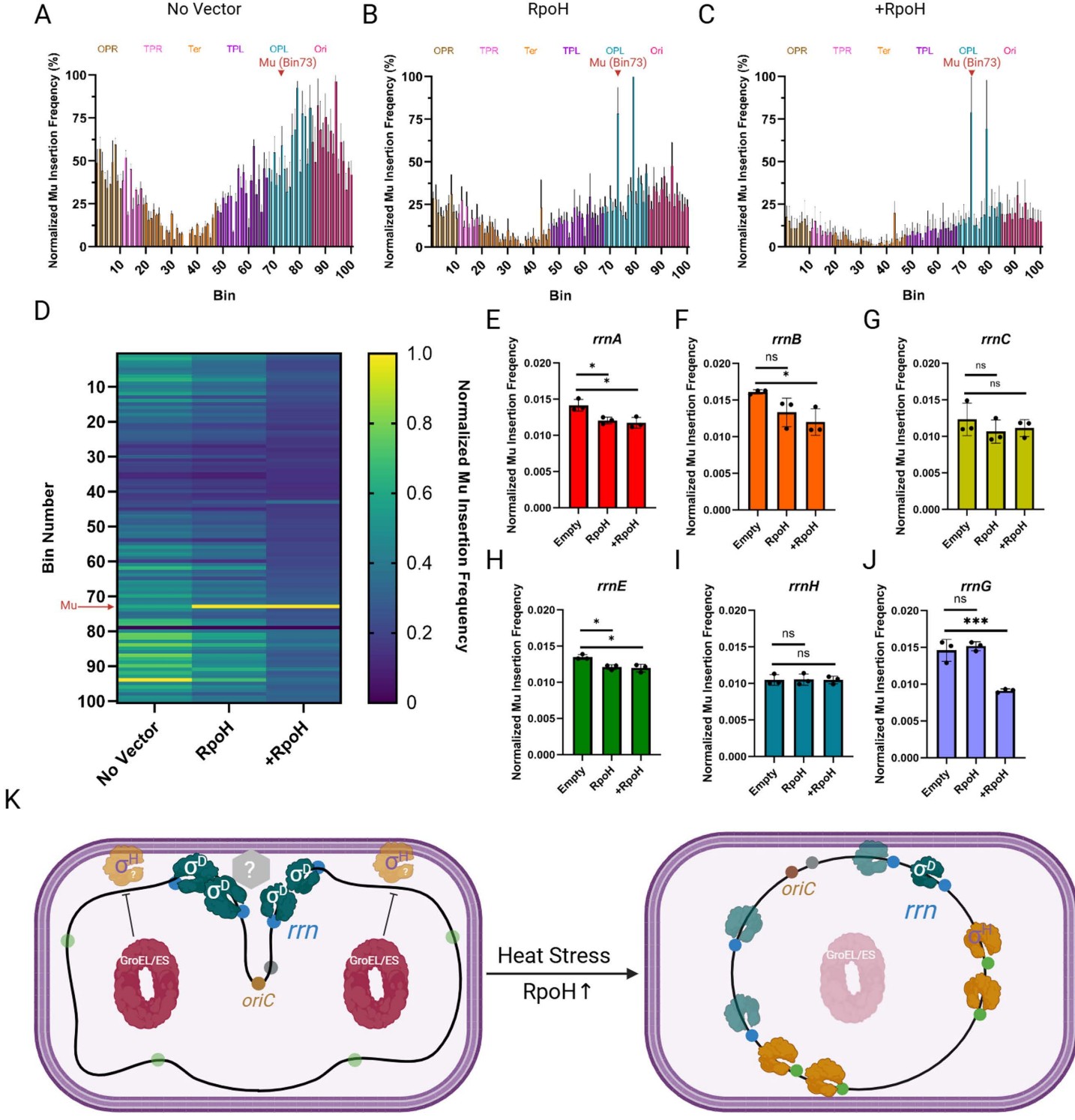

**Fig 4. Expression of *rpoH* disrupts *rrn* clustering as tracked by Mu transposition. (A)** Frequency of transposition of Mu located in the vicinity of *rrnD* (Bin 73) after one round of transposition. The number of insertions has been normalized to the read depth of each bin but not normalized to copy number. In the asynchronous population, many cells have partially replicated chromosomes [8]. The initial Mu position is indicated by a red triangle. The *Escherichia coli* genome was partitioned into 100 equally sized bins, so each bin is ~46 kb. Starting bin number and chromosomal regions are indicated on top. Each vertical bar represents the average normalized transposition frequency of three biological replicates at the indicated bin, expressed as a

percentage (with the highest transposition frequency being set to 100%). Gray error bars are the standard deviation. Color of bars correspond to regions of the *E. coli* chromosome annotated up top [8]. **(B)** Same as (A) but with the *rpoH* vector. **(C)** Same as (B) but with induction of *rpoH*. **(D)** Heat map of the data from A through C. In the no-vector control, the highest insertion frequency (yellow bar) occurs in Bin 94 (close to *oriC* which resides in Bin 90). **(E–J)** Transposition frequency from *mlaF* (*rrnD*) into the indicated *rrn* operon-containing bin under conditions described in A through C. The individual data points and associated standard deviations are shown. Statistical significance was determined with Student *t* test (two-tailed), *$p < 0.05$. ns: not statistically significant. **(K)** Model for σ$^D$-mediated *rrn* clustering. σ$^D$-bound RNAP (dark green) is responsible for clustering *rrn*s (blue circles) at the membrane either directly or through other unknown factor(s) (gray hexagon). GroEL/ES (red oval) represses the activity of σ$^H$ (orange) preventing it from competing with σ$^D$ for core RNAP. σ$^H$ is typically localized to the inner membrane and degraded. Upon heat stress, σ$^H$ levels rise, and GroEL/ES, being diverted to un-folded substrates (pink oval), is unable to repress σ$^H$ activity. Elevated σ$^H$ levels from a plasmid, phenocopy the heat response. σ$^D$-bound RNAP complexes are displaced through competition with increased σ$^H$, and σ$^H$-bound RNAP now transcribes genes in σ$^H$ regulon (green circles), lowering the total amount of RNAP transcribing *rrn* operons at the membrane, destabilizing clustering. Data used for plotting Fig 4 graphs can be found in the Supplemental "S1 Data" file under appropriate figure headings. Created in BioRender. Ho, K. (2025) https://BioRender.com/d36h492.

methodologies—FROS and Mu transposition [6,8]. Gaal and colleagues (FROS methodology) called this organization a "bacterial nucleolus" and showed that its presence was independent of growth media and cell doubling times [6]. Although the stronger of the two promoters, P1, and it UP element in *rrnD*, were required, active *rrn* transcription was ruled out because co-localization of the loci queried was resistant to disruption by rifampicin, which stops RNA chain elongation [38], as well to mutation of a −10 region of the P1 promoter required for initiation of transcription. These properties of the *E. coli* nucleolus are in contrast to the requirement for Pol I function and rRNA transcription to maintain nucleolar structure and integrity in eukaryotic cells [39]. We will therefore simply call this co-localization a *rrn* cluster.

To learn more about the nature of the *rrn* cluster, we attempted to perturb it by exposing cells to several environmental stressors (Fig 1B). Of these, heat and ethanol shock had the largest effect. These two stressors share a common "unfolded protein" response [11]. The important takeaway from this experiment is that by perturbing the cluster, we had not only validated its existence but found a handle to probe its nature.

**The heat shock response, GroEL/GroES and σ$^H$ all destabilize the *rrn* cluster: σ$^D$ counteracts the action of σ$^H$**

Adaptation to heat shock is a universal biological phenomenon [11]. Heat denatures proteins, so organisms adapt by synthesizing protein-folding chaperones. *E. coli* encodes several chaperones including GroEL/GroES and employs σ$^H$ to transcribe hundreds of genes that enable bacterial survival [13,14]. The ability of heat shock to disrupt the *rrn* cluster (Fig 1B–1D) suggested to us that proteins must participate in holding the structure together at some level. We therefore directed FLAG-tagged dCas9 immediately upstream of each of the 7 *rrn* loci, followed by pulldown with FLAG antibodies and mass spectroscopy. No common protein was significantly enriched across all samples, but GroEL was enriched in 4/7 sgRNAs (*rrnC*, *rrnA*, *rrnB*, and *rrnD*), suggesting that GroEL could be a possible *trans*-acting factor in clustering (S4B, S6, and S7 Figs). (Although *rrnC* is not part of the cluster, it is close to the origin of replication *ori*, and it is conceivable that unrelated events at *ori* block its incorporation into the cluster).

The *groEL/S* operon is transcribed exclusively by σ$^H$, whose levels are kept low through multiple mechanisms, including sequestration at the inner membrane where it is degraded by protease FtsH, as well as direct interaction with GroEL/ES [27,28]. Attempts to query the participation of GroEL by providing it either from regulated promoter or its native promoter resulted in disruption of the *rrn* cluster (S4C–S4D Fig). To test if our manipulations of GroEL were perturbing σ$^H$ levels, we provided RpoH ectopically from a regulatable promoter. Even in the absence of induction, leaky expression of RpoH was sufficient to disrupt the *rrn* cluster (Fig 2A).

Why should an apparently slight rise in σ$^H$ levels have such a profound effect on stability of the cluster? Given that the P1 promoter and its UP elements somehow participate in maintaining the cluster, we imagined a scenario where RNAP bound to the housekeeping σ$^D$, known to transcribe the *rrn* operons [5], was stationed there, and that σ$^H$ might be competing with it for binding RNAP, displacing it and disrupting the structure. Testing this conjecture by inserting *rpoD* along with *rpoH* on the ectopic vector resulted in significant cluster rescue (Fig 2A), supporting our conjecture.

**Sigma factors that disrupt the *rrn* cluster associate with the membrane, suggesting a model for *rrn* clustering**

Competition among sigma factors for the core RNAP has been proposed to be the mechanism for gene regulation [31]. If such a competition model applies to why rise in σ$^H$ disrupts the cluster, then mutants of σ$^H$ that are defective for either RNAP-core or DNA-binding should not disrupt the cluster. RpoH mutants defective for interaction with RNAP (L245P) and for promoter binding at the −35 region (E265A), were both significantly deficient in the de-clustering activity of WT RpoH (Fig 2C), in keeping with the competition model.

σ$^H$ levels are regulated by at least three mechanisms: control through GroEL/GroES, DnaJ/K/GrpE, control of translation efficiency at the mRNA level, and the degradation of σ$^H$ by FtsH, an integral-membrane protease. A third RpoH mutant we tested for its de-clustering activity was I54N, shown to escape FtsH-mediated proteolysis. Our expectation was that this mutant would behave like WT RpoH. Despite its high transcription activity (Fig 2D), however, the effect of the I54N mutant did not align with our expectations (Fig 2C), showing at the very least that high levels of transcription directed by σ$^H$ are not the cause of de-clustering, and that none of the members of the σ$^H$ regulon are involved in de-clustering. So why did the I54N mutant not behave like WT RpoH? Unlike the WT protein, I54N mutant is unable to localize to the membrane, suggesting that access to the membrane is important for the de-clustering effect of RpoH, ergo, the *rrn* cluster might be anchored in the membrane.

Of the five other sigma factors tested for cluster-disrupting ability, only FecI, an extracytoplasmic function (ECF) σ factor, also disrupted clustering, although not as severely as σ$^H$ (Fig 3D). FecI, along with FecA and FecR, is responsible for the transcription of the ferric citrate transport system, consisting of *fecABCDE* transport genes [40]. FecA is an outer membrane protein that transports $(Fe^{3+}\text{-citrate})_2$ across the outer membrane. Upon binding of $(Fe^{3+}\text{-citrate})_2$ to FecA, the signal is then transduced through the periplasmic face of FecR, an inner membrane protein, to its cytoplasmic face [41,42]. This conformational change of FecR activates the transcriptional activity of FecI in the cytoplasm and promotes transcription of *fecABCDE* [43]. The dependence of FecI on FecR, which is localized to the inner-membrane, for efficient transcription by RNAP-FecI of the *fec* operons, also lends support to a model where the site of *rrn* clustering is the inner membrane. Why then did overexpression of σ$^E$, which is also an ECF, fail to disrupt clustering [44,45]? The mode of regulation of σ$^E$ is based on the repression of σ$^E$ by the inner membrane anti-sigma factor RseA, inhibiting the interaction of RNAP core and σ$^E$ [46,47]. Upon membrane stress, such as ethanol or heat, DegS is activated by the presence of unfolded outer-membrane proteins, and, in concert with RseP, degrades RseA, releasing σ$^E$ to promote transcription of downstream genes [48,49]. The negative regulation by RseA may explain why overexpression of σ$^E$ alone is unable to disrupt clustering. In other words, for efficient disruption of the *rrn* cluster, σ$^D$-independent transcription needs to occur at the inner membrane.

Taken together, we propose that clustering of *rrn* is mediated by σ$^D$ at the membrane either through σ$^D$ directly or through other unknown factor(s) (Fig 4K). σ$^D$ has indeed been observed at the membrane [28]. Given that high levels of transcription are not required, but the *cis*-acting elements responsible for high transcriptional activity of *rrn* are, and that some σ factors can disrupt clustering implicating σ$^D$ as the clustering factor, we propose that the assembly of RNAP holoenzyme and UP element features of P1 drives *rrn* clustering. Upon heat stress or elevated σ$^H$ levels, RNAP complexes bound to σ$^D$ are displaced through competition with σ$^H$, and to a lesser extent FecI, for RNAP core, resulting in de-clustering of *rrn* operons. The order of RNAP assembly has been shown to be $\alpha_2 \rightarrow \alpha_2\beta \rightarrow \alpha_2\beta\beta'\omega \rightarrow \alpha_2\beta\beta'\omega\sigma$ [50]. The requirement for the P1 promoter may reflect the high affinity of α for the UP element present upstream of the −35 element [51]. The dimerization of the α subunit could drive the bridging interaction between the disparate *rrn* operons. However, this interaction by itself is unable to explain *rrn* clustering as demonstrated by de-clustering effect of σ$^H$ and FecI, since σ factor binding and unwinding of the DNA duplex is the last step of RNAP assembly. Therefore, we propose that two elements are responsible for *rrn* clustering: (1) the high affinity of α for the UP element, and (2) DNA binding driven by σ$^D$. The affinity of α for the *rrn* UP element would explain why *rrn* clustering does not extend to other non-*rrn* loci. The specificity of σ$^D$ for the *rrn* promoter would explain the de-clustering activity of σ$^H$ and FecI. We note that σ$^H$ did not affect the position of two pairs that shared *rrnH* (Fig 3C),

suggesting a likely sub-organization within this structure. Finally, why did we recover GroEL at 4/7 rDNA loci (S4A–S4C, S6, and S7 Figs)? One possibility is that GroEL is meant to clear σ$^H$ from the cluster [27]. Alternatively, since GroEL has been shown to restore transcription of heat-treated RNAP in vitro [52], it could either be interacting non-specifically with some component of RNAP or ensuring that the proteins at the cluster do not aggregate.

### Relationship of the *rrn* cluster to RNAP condensates

RNAP has been reported to form distinct clusters on the *E. coli* nucleoid [53,54]. Since the bulk of cellular transcription is dedicated to rRNA transcription and since RNAP clusters form in fast-growth conditions, it was proposed that these clusters represent high concentrations of RNAP on *rrn* operons [55]. Subsequent work showed that RNAP clusters colocalize with nascent rRNA, but that their spatial arrangement was not dependent on rRNA synthesis activity and was likely organized by the underlying nucleoid [56]. RNAP clusters have been shown to be biomolecular condensates capable of phase separation, involving known factors associated with RNAP (i.e., ω subunit of RNAP and NusB) [21].

The properties of the *rrn* cluster are on the one hand reminiscent of RNAP condensates in that high level of transcription of rRNA is not required for their organization, but, on the other hand, are different in that the cluster is immune to transcription inhibitors while the condensates are not [56]. The cluster is unlikely to serve as a precursor for formation of RNAP condensates since these form in a strain where only one *rrn* operon is present [48]. Antitermination factors such as NusB and ω subunit of RNAP have been shown to contribute to formation of RNAP condensates [23]. However, we showed in this study that neither Δ*nusB* nor Δ*rpoZ* strains significantly impact the cluster (S3 Fig). We interpret these results to mean that RNAP condensates represent a feature of highly active transcription that the *rrn* cluster contributes to by making rDNA readily available through spatial localization.

## Materials and methods

### Media, strains, phages, and plasmids

Unless conditions are specified, all strains are grown in LB at 30 °C with shaking. When appropriate, antibiotics were at the following concentrations: Ampicillin (Amp) at 100 µg/mL, Kanamycin (Kan) at 25 µg/mL, Chloramphenicol (Cam) at 20 µg/mL. Anhydrotetracycline (aTc) was used for induction of Tet promoter at 50 ng/mL. Isopropyl-β-D-Galactoside (IPTG) was used to induce the *lac* promoter at 1 mM. O-nitrophenyl-β-D-Galactoside (ONPG) was purchased from Sigma. Competent cells for transformation were prepared by washing a growing culture of O.D. 0.4–0.5 in cold 10% glycerol three times. The pellet was resuspended in 1:100 of the original volume in 10% glycerol. Electroporation was performed in *E. coli* Pulser (Biorad) with 1 mm Electroporation cuvette Plus (Fisher) at 1.8 V. Cells were recovered in SOC (LB supplemented with 10 mM MgCl$_2$, 10 mM MgSO$_4$, and 0.2% Glucose) for 1.5 h at 30 °C with shaking prior to plating on the appropriate selection. Mu phage was stored in Mu Buffer (50 mM Tris-HCl pH 8.0, 100 mM NaCl, 5 mM CaCl$_2$, 5 mM MgCl$_2$, and 0.1% gelatin). Strains and Phages employed in this study are listed in S1 Table. Primers, purchased from Integrated DNA Technology, used in this study are listed in S2 Table.

### General molecular techniques

Routine PCR was performed with Taq DNA Polymerase (NEB), according to manufacturer's instructions. PCR fragments for cloning were generated with Phusion DNA Polymerase (NEB) according to manufacturer's instructions. Gibson Master Mix was made according to Gibson *and colleagues* [57]. Gibson assembly was performed at 50 °C for at least 2 h. Golden Gate Assembly was performed with Esp3I (NEB), T7 DNA Ligase (NEB), and T4 DNA Ligase Buffer (NEB). Primers used to generate gRNAs were used at 10 nM each. Hundred nanograms (100 ng) of the destination vector (pMAZ-SK sfgfp) was used. The program for Golden Gate Assembly was 3 min at 37 °C, 2 min at 16 °C for 35 cycles, followed by 1 cycle of 10 min incubation at 37 °C. T4 DNA Ligase, T4 Polynucleotide kinase (PNK), and T4 DNA Ligase Buffer were purchased from NEB.

## Plasmid construction

Plasmids were constructed using Gibson assembly. Typically, 250 μL of the backbone was assembled with a molar equivalent of insert in a 20 μL reaction. The resulting product was purified with PCR clean-up kit (Qiagen) according to the manufacturer's instructions. Two microliters (2 μL) of the 20 μL eluted product was used for transformation. Colonies were screened by PCR with primers spanning the junction between the backbone and the insert. Positive clones were restruck and checked once more using the same primer pairs. The PCR product was sequenced to confirm the identity of the sequence at either the UT Core Sequencing Facilities or Eton Biosciences. For generating single-point mutants, primers carrying the desired mutation were used to amplify the plasmid of interest. The resulting PCR product was gel-extracted with Qiagen Gel Extraction Kit. The product was then self-ligated overnight with T4 DNA Ligase and T4 DNA PNK in T4 Ligase Buffer. The ligated product was then purified and 2 μL of the 20 μL eluted product was used for transformation. Positive clones carrying the desired mutation were identified by PCR of the target sequence followed by sequencing. The positive clones were then restruck once again and verified with PCR and sequencing.

## Strain construction

For insertion of *parS* sequences, the procedure was essentially as described in [58] with the exception being the template plasmid (pKH3 or pKH4) carrying the appropriate *parS* linked to antibiotic resistance cassette. Briefly, 0.5 mL of an overnight culture of MG1655 carrying pKD46 was pelleted, washed twice in 1 mL of phosphate-buffered saline (PBS), and diluted 1:100 in fresh LB supplemented with 0.2% arabinose. The culture was grown to an O.D. of approximately 0.4. The cells were made electrocompetent. Cells were then transformed with the appropriate PCR product and let recover in SOC for 3 h at 30 °C. The outgrowth was then plated on the appropriate selection and incubated at 37 °C overnight. Positive clones were identified by PCR with primers amplifying the junction of the expected insertion. Positive clones were struck out on the appropriate selection plate at 37 °C and reconfirmed with PCR followed by sequencing. To remove the antibiotic-encoding cassette for subsequent insertion of additional *parS*, pCP20 was transformed into the desired host strain. Clones carrying pCP20 were then struck out on LB plate without selection and incubated at 42 °C overnight. Colonies were then checked for the loss of both pCP20 and the antibiotics cassette by streaking on the appropriate selection.

## Fluorescent microscopy and post-processing

Overnight cultures used for fluorescent microscopy were grown overnight in EZ-Rich media (Teknova) from a single colony and diluted 1:100 in fresh EZ-Rich media supplemented with IPTG for induction of ParB-fluorescent fusions from pFHC2973. The subculture grew until an O.D. of 0.4. One milliliter of the culture was then pelleted by centrifugation and resuspended in 100 μL of PBS. Six microliters of the suspension was then spotted onto agarose pad (1%) and let dry. The sample was then observed under Olympus-BX 53 microscope, XM10 monochrome camera on the 100× objective with oil immersion. Most images were taken at an exposure of ~100 ms for GFP filter, and ~400 ms for CFP filter, however, some samples required a longer exposure time to obtain acceptable signal for downstream processing; the upper limit of exposure was 3 s. Cellular stressors were applied 20 min prior to imaging, after which they were prepared and imaged as described above. For heat stress, the cells were transferred to a 42 °C water bath. SHX was added to a concentration of 50 μM. Cold stress was induced by incubating the cells in a 4 °C water bath. Ethanol stress was induced by adding ethanol to 0.5%. For image processing, the background was subtracted from the image, and foci were detected with ImageJ using the detect maxima function. The coordinates of the foci were exported and the Cartesian distance between a GFP focus, and its closest CFP focus was determined with custom Python script. The distance was computed by scaling the distance in pixel to nm with 1 pixel = 62 nm. Cell length was measured with ImageJ using the same scale as described above (1 pixel = 62 nm). For each experiment, three biological replicates were examined, and one representative is shown. Code used for analyzing the foci can be found at https://doi.org/10.5281/zenodo.14927315.

## DAPI Staining and nucleoid measurements

Cells were grown and subjected to heat stress or *rpoH* expression in manner identical that used for microscopy. One milliliter of cells were then washed twice in PBS and resuspended in 200 μL of diH$_2$O. Thirty-seven percent Formaldehyde (Fisher) was added to a final concentration of 3.5% and incubated at 4 °C for 30 min, after which time, the cells were washed three times in 1 mL of PBS and resuspended in DAPI solution (500 nM in diH$_2$O, 200 μL) for 10 min. The cells were then washed once in 1 mL of PBS, resuspended in 100 μL of PBS, mounted for microscopy, and visualized with DAPI filter. The signal intensity for DAPI was measured by ImageJ.

## Pulldown of dCas9 and proteomics

Three independent overnight cultures of MG1655 carrying pKH5 and the corresponding gRNA were pelleted and washed as described above. The pellet was diluted 1:100 into 100 mL fresh LB supplemented with selection and 0.2% arabinose and aTc. The cultures were grown to an O.D. of 0.6 and pelleted and washed three times in 1 mL of PBS. The subsequent pulldown procedure was carried out as described by FLAG Immunoprecipitation kit (Rockland). Every step of the pulldown was conducted at 4 °C. Briefly, the pellet was resuspended in 5 mL of lysis buffer and sonicated (Brason tip, 40% intensity) for 10 min (10 s on, 10 s off cycle). The lysate was clarified by centrifugation at 4 °C. Agarose-αFLAG Ab was washed twice in PBS and once in elution buffer. The washed Agarose-αFLAG Ab was incubated with the lysate overnight. After incubation, the beads were collected and washed 3 times with PBS. The bound proteins were eluted with elution buffer. The proteins were quantified by Mass Spectrometry at the UT Proteomics core. Samples were digested with trypsin, desalted and run on Dionex LC (liquid chromatography) and Orbitrap Fusion 2 (mass spec machine) for 60 min. Raw data were analyzed with PD2.2 and Scaffold 5 software. Downstream analysis was performed as described in [23] with *lacZ* gRNA pulldown as the negative control. Significantly enriched hits were ranked based on a z-score cut-off of 2.5.

## Miller assay

0.5 mL of three overnight cultures of the desired strain carrying the appropriate plasmid was pelleted and washed in 1 mL of PBS twice. The pellet was then resuspended in 0.5 mL of PBS and diluted 1:100 in LB. The cultures were grown to an O.D. of 0.4 and aTc was added. At indicated time points, 20 μL of the culture was withdrawn, its O.D. 600 recorded and added to 80 μL of permeabilization buffer (100 mM NaHPO$_4$, 20 mM KCl, 2 mM MgSO$_4$, 0.8 mg/mL CTAB, 0.4 mg/mL sodium deoxycholate, 5.4 μL/mL β-mercaptoethanol). After the final time point, 600 μL of substrate solution (60 mM Na$_2$HPO$_4$, 40 mM NaH$_2$PO$_4$, 1 mg/mL ONPG, 2.7 μL/mL β-mercaptoethanol) was added to each sample. The samples were incubated at 30 °C for 60 min before 600 μL of 1M Na$_2$CO$_3$ was added to stop the reaction. O.D. 420 was recorded. Miller units were calculated as follows:

$$Miller\ Units = 1000 \times \frac{Absorbance_{420nm}}{60 \times Absorbance_{600nm} \times 0.02}$$

## Mu phage preparation

When required, a Mu prophage strain was grown overnight, diluted 1:100 in fresh LB and grown until an O.D. of 0.5. After which, the culture was shifted to a 42 °C water bath and incubated until lysis is complete. The lysate was clarified by centrifugation at 6000$g$ for 20 min. The supernatant was transferred to a clean flask and NaCl was added to a final concentration of 0.5M followed by the addition of Polyethylene Glycol 8,000 to a final concentration of 10% w/v. The mixture was incubated overnight at 4 °C. The pellet was then collected by centrifugation at 8000$g$ for 20 min. The pellet was then resuspended in 1:100 of the original volume in Mu Buffer. Chloroform was added to the mixture and shaken. The phases

were separated by centrifugation at 4000*g* for 10 min at 4 °C. The aqueous phase was collected (top layer) and phage titer determined prior to use.

### Generation of mlaF::Mu and single-transposition experiments

Selection for Mu insertion at *mlaF* (bin72 as described in [8]) was performed by first introducing *sacB* at *mlaF* followed by infection Mu carrying a Cam marker. Briefly, *attP* was introduced into *mlaF* by λred recombination, and the Kan marker was removed. KH2 was then introduced into this strain, induced for expression of λ Int with arabinose followed by transformation of KH22. Clones positive for integration of the *sacB-kanR* cassette was verified by PCR spanning the expected junction between the cassette and *mlaF*. This strain was then infected with Mu::Cam at an MOI (multiplicity of infection) of 5, and survivors that were resistant to sucrose and Cam were selected for. Three colonies were picked and insertion locations for each were confirmed with PCR. Southern Blot was performed to confirm that only one Mu inserted at the desired location. Of three clones picked, one was confirmed to be a mono-lysogen. This strain was designated KH10. Single hop experiments with KH10 were performed as described in [8] with modifications for plasmid expression of *rpoH*. For induction of *rpoH* in KH10, overnight cultures were selected for Tet and subsequently grown in LB absent for Tet. One hour prior to temperature shift for induction of Mu transposition, aTc was added. Genomic DNA was extracted with Wizards Genomic DNA Kit, and sequencing was performed by Novogene on NovaSeq PE150 platform. Partitioning of the genome into 100 bins was described previously [8]. Data was processed using custom script described in [8] without LASSO regression for normalization. Instead, the number of insertions per bin was first corrected for total number of read counts, followed by correcting for replication effect by normalizing to the total number of *E. coli* reads mapped to that bin. The average of three biological replicates and standard deviation was used to plot data for normalized transposition frequency.

### Supporting information

**S1 Fig. Distribution of rrn operons on the *Escherichia coli* genome.** On the circular *E. coli* chromosome, replication originates at oriC, with two bidirectional replication forks traversing each arm (replicore), terminating within the Ter region. Locations of the rrn operons shown to cluster by two different methodologies (see main text) are indicated by blue-filled circles. Positions of the operons on the circular chromosome are not drawn to scale. Organization of a typical rrn operon is shown for rrnG, with tRNAGlu2 between 16S and 23S; some rrn loci possess distal tRNAs. P1 and P2 promoters are indicated by forward arrows. Created in BioRender. Ho, K. (2025) https://BioRender.com/g85e152.
(TIFF)

**S2 Fig. Microscopy of cells labeled with *parS*-ParB at *rrnA* and *rrnD*.** (A) Insets of a complete field of cells (phase contrast and fluorescence) shown in B. *rrnA* and *rrnD* were tagged with GFP-ParB and CFP-ParB, respectively, as described in Fig 1A, and processed as described in Fig 1B. (B) Complete field and position of identified foci. Created in BioRender. Ho, K. (2025) https://BioRender.com/p20w223.
(TIFF)

**S3 Fig. Effect of Δ*nusB*, Δ*rpoZ*, and Δ*nc5* on the distance between *rrnA* and *rrnD*.** Effect of deletions of factors known to participate in RNAP condensate on *rrnA-rrnD* distance. Experimental conditions are similar to those in Fig 1. Data used for plotting S3 Fig graphs can be found in the Supplemental "S1 Data" file under appropriate figure headings. Created in BioRender. Ho, K. (2025) https://BioRender.com/a48g408.
(TIFF)

**S4 Fig. Deregulation of GroEL disrupts *rrnA-rrnD* clustering.** (A) Scheme for pulldown of proteins in the vicinity of all rrn loci. dCas9-FLAG (pink/green) was directed upstream of all 7 rrn loci by expressing sgRNA specific for each target. Formaldehyde was used to crosslink dCas9 to putative bridging factor(s) (purple). The dCas9-linked 'complex' was

then immunoprecipitated and subjected to mass spectrometry. (B) Significantly enriched proteins in pulldown with gRNA targeting yeiP (rrnC). Each protein is identified by a circle. Each axis represents the $z$-score of each protein in separate experiments. Lines from the axes indicate the cut-off for enrichment. Red circles and black circles indicate proteins that fall above and below the False Discovery Rate (5%), respectively. Proteins significantly enriched ($z$-score>2.5) are in the top-right square (red balls with corresponding protein name). See S6 Fig for data obtained for the remaining gRNAs. (C) Distance between rrnA-D operons in WT and ΔgroEL strains expressing either both groES/groEL or groEL alone from pAraBAD plasmid. Other descriptions as in Fig 1B. (D) Distance between rrnA-D in WT and ΔgroEL strains expressing groES/groEL from either pAra or the native groE promoter. Data used for plotting S4 Fig graphs can be found in Supplemental "S1 Data" file under appropriate figure headings. Created in BioRender. Ho, K. (2025) https://BioRender.com/z72h116.
(TIFF)

**S5 Fig. Efficiency of gRNA in cell lethality. gRNAs employed in S4 Fig were assayed for their ability to target chromosomal loci by substituting dCas9 with Cas9 and monitoring resultant cell death.** Lethality was determined by the fraction of CFU that survive induction of Cas9 and gRNA compared to the CFU of non-induced control (y-axis). Gene names on the x-axis indicate the targeted gene immediately upstream of the rrn operon indicated in parentheses. gRNA targeting lacZ without (bolded) and with the protospacer adjacent motif, were used as negative and positive controls, respectively. Data used for plotting S5 Fig graphs can be found in Supplemental "S1 Data" file under appropriate figure headings.
(TIFF)

**S6 Fig. Proteins enriched in gRNA pulldowns.** (A–F) Results from gRNAs directed to indicated rrn loci. See S4A Fig legend for other experimental details. Data used for plotting S6 Fig graphs for each gRNA pulldown can be found in S3–S9 Tables. Created in BioRender. Ho, K. (2025) https://BioRender.com/i51m760.
(TIFF)

**S7 Fig. Upset plot of the number of proteins identified in each gRNA pulldown.** Aggregated MS results of significantly enriched proteins from dCas9 pulldown. The vertical bar graph on the top shows the number of proteins identified for each individual pulldown (black balls). The first seven data points are unique to each sgRNA targeted *rrn*. The horizontal bars indicate the relative protein numbers in each of the unique pulldowns. The black lines connecting the other black balls indicate co-occurrence of the proteins in individual pulldowns. Data used for plotting S7 Fig graph for each gRNA pulldown can be found in S3–S9 Tables. Created in BioRender. Ho, K. (2025) https://BioRender.com/z91f416.
(TIFF)

**S8 Fig. Growth of ΔgroEL strain complemented of groEL from pAraBAD.** Colony morphologies upon induction and repression of GroEL with added arabinose or glucose, respectively, at 30 °C (A) and 37 °C (B). See S4D Fig and related text. Created in BioRender. Ho, K. (2025) https://BioRender.com/t72u237.
(TIFF)

**S9 Fig. RpoH disrupts *rrnA-rrnD* clustering.** Representative image of *rrnA-rrnD* foci showing no loss of *rrnA*-GFP foci as seen with heat stress in Fig 1D. Created in BioRender. Ho, K. (2025) https://BioRender.com/t54z191.
(TIFF)

**S10 Fig. Effect of heat stress and *rpoH* overexpression on the nucleoid and the distance between non-*rrn* loci.**
(A) Cell length distribution under heat stress and *rpoH* overexpression. Conditions for heat stress and *rpoH* overexpression are identical to that of Figs 1B and 3A, respectively. 200–300 cells were measured for each sample. Data shown are

pooled from three biological replicates. (B) Condensation of the nucleoid under the same conditions as in A, with similar cell numbers and replicates. (C) Conditions tested in A do not affect the distance between *lacZ* and *leuC*. Left, relative position of *lacZ* and *leuC* on the genetic map (not drawn to scale). Right, spatial distance (120 nm, 120 nm, and 85 nm for No Vector, RpoH, and +RpoH, respectively) between *lacZ*-pMT*parS* and *leuC*-P1*parS* as measured by fluorescent microscopy. Conditions for *rpoH* overexpression and induction are identical to that of Fig 3A. Data used for plotting S5 Fig graphs can be found in Supplemental "S1 Data" file under appropriate figure headings. Created in BioRender. Ho, K. (2025) https://BioRender.com/a87h018.
(TIFF)

**S11 Fig. Morphology of cells over-expressing *rpoH* variants.** Phase contrast images of cells carrying the indicated mutation in *rpoH* were grown, imaged, and induced as described in Methods. Created in BioRender. Ho, K. (2025) https://BioRender.com/w03v569.
(TIFF)

**S1 Table. Strains and plasmids used in this study.**
(DOCX)

**S2 Table. Primers used in this study.**
(DOCX)

**S3 Table. Spectral data of gRNA targeting *rrnA* pulldown.**
(CSV)

**S4 Table. Spectral data of gRNA targeting *rrnB* pulldown.**
(CSV)

**S5 Table. Spectral data of gRNA targeting *rrnG* pulldown.**
(CSV)

**S6 Table. Spectral data of gRNA targeting *rrnD* pulldown.**
(CSV)

**S7 Table. Spectral data of gRNA targeting *rrnE* pulldown.**
(CSV)

**S8 Table. Spectral data of gRNA targeting *rrnH* pulldown.**
(CSV)

**S9 Table. Spectral data of gRNA targeting *rrnC* pulldown.**
(CSV)

**S1 Data. Raw data for figures that contain numerical data.** The name on each tab indicates the figure for which the data were used to plot.
(XLSX)

## Acknowledgments

We thank Rick Gourse for providing pFHC2973 and *parS*-labeled *rrn* strains, Ian Molineux for BW25141 for propagation of R6K origin plasmids, Kamyab Javanmardi for providing Golden Gate vector for gRNA insertion, Rachael Cox and Edward Marcotte for script and computing power used in mass spectrometry analysis, Brady Wilkins for assistance with Southern Blot, and Lydia Freddolino for comments on the manuscript. Figures were created with Biorender.

## Author contributions

**Conceptualization:** Khang Ho, Rasika M. Harshey.

**Data curation:** Khang Ho.

**Formal analysis:** Khang Ho.

**Funding acquisition:** Rasika M. Harshey.

**Investigation:** Khang Ho.

**Methodology:** Khang Ho, Rasika M. Harshey.

**Project administration:** Rasika M. Harshey.

**Software:** Khang Ho.

**Supervision:** Rasika M. Harshey.

**Validation:** Khang Ho.

**Visualization:** Khang Ho.

**Writing – original draft:** Khang Ho, Rasika M. Harshey.

**Writing – review & editing:** Khang Ho, Rasika M. Harshey.

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
