## [Editor Report · Decision Letter 0]

1 Oct 2024

Dear Dr Harshey,

Thank you for submitting your manuscript entitled "Clustering of rRNA operons in E. coli is disrupted by σH" for consideration as a Research Article by PLOS Biology. Please accept my sincere apologies for the delay in getting back to you with feedback as we consulted with an academic editor about your submission.

Your manuscript has now been evaluated by the PLOS Biology editorial staff, as well as by an academic editor with relevant expertise, and I am writing to let you know that we would like to send your submission out for external peer review.

Once your full submission is complete, your paper will undergo a series of checks in preparation for peer review. After your manuscript has passed the checks it will be sent out for review. To provide the metadata for your submission, please Login to Editorial Manager (https://www.editorialmanager.com/pbiology) within two working days, i.e. by Oct 03 2024 11:59PM.

Kind regards,

Richard

Richard Hodge, PhD

rhodge@plos.org

PLOS

---

## [Decision Letter · Decision Letter 1]

6 Nov 2024

Dear Dr Harshey,

Thank you for your patience while your manuscript "Clustering of rRNA operons in E. coli is disrupted by σH" was peer-reviewed at PLOS Biology as a Research Article. Please accept my sincere apologies for the delays that you have experienced during the peer review process. Your manuscript has now been evaluated by the PLOS Biology editors, an Academic Editor with relevant expertise, and by three independent reviewers.

In light of the reviews, which you will find at the end of this email, we would like to invite you to revise the work to thoroughly address the reviewers' reports.

As you will see below, the reviewers are positive about your manuscript and think the results are interesting and well done. However, the reviewers raise some overlapping concerns, such as the design of the dCas9 pulldown assays and the specific role of RpoH in rrn operon clustering. In addition, Reviewer #1 asks about the measures taken to ensure reliable comparisons across the variable stressors, as well as controls for the microscopy data.

Given the extent of revision needed, we cannot make a decision about publication until we have seen the revised manuscript and your response to the reviewers' comments. Your revised manuscript is likely to be sent for further evaluation by all or a subset of the reviewers.

**IMPORTANT - SUBMITTING YOUR REVISION**

*Re-submission Checklist*

*Published Peer Review*

*PLOS Data Policy*

*Blot and Gel Data Policy*

Sincerely,

Richard

Richard Hodge, PhD

rhodge@plos.org

REVIEWS:

Reviewer #1: In a previous study, the group of Prof. Harshey provided evidence for long-range interactions on the E. coli chromosome. Using a Mu-based transposition assay, they found that rrn loci clustered together, despite being located across the E. coli chromosome. They identified some DNA elements in the region of these loci as being essential for the clustering. In the present study, Ho and Harshey probe for proteins that may mediate this clustering. For this, they first screen for stress conditions where the clustering is disrupted. They identify ethanol and heat-shock stresses as two conditions where clustering is reduced. Next, they carry out a pull down (via dCasd9 targeting) of the clustered rrn operon regions. They find that GroEL is common across 4 of the rrn operons. Further probing reveals that perturbation in the GroEL-GroES ratios affects clustering, and this may be driven by sigma factor competition. The authors go on to shown that sigmaH (expressed during heat shock) can disrupt clustering, likely via competition with sigmaD. Mutational analysis suggests that this is not dependent on active transcription but relies on sigmaH association with DNA and its RNA polymerase interactions.

The study is interesting as it probes into the mechanism of rrn operon clustering in E. coli. Given the limited examples of such clustering in bacterial genomes, the findings will be of broad relevance to the chromosome biology community across bacteria and eukaryotic systems. I do have some points that authors should consider.

1. Comparison across diverse stressors. Authors use a diverse variety of stressors to assess which conditions affect rrn clustering. However, comparison across these conditions is challenging as they may not affect cell growth to the same extent. It is unclear what concentrations/ range of stress conditions were tried to enable reliable comparison across stringent response, cold shock, ethanol and heat shock.

2. Microscopy observations:

a. In all stress conditions, cell lengths and global chromosome condensation/ organization is subject to change when compared to wildtype. Hence it is insufficient to measure inter-locus distance between rrn loci alone. Authors should provide cell length distributions and DAPI staining analysis in all imaged conditions to show that cell lengths and chromosome compaction are comparable to wild type, even though the inter-locus distance is changing.

b. In addition, authors should measure the inter-locus distance between two chromosome loci known to not cluster, as a control to show that the sigma factor effects are specific to the rrn operons.

c. It is unclear what microscope was used for the imaging (details are missing in the methods). If this is widefield imaging, do the authors have the resolution power to differentiate across 216-230nm? In my experience any localizations below 250nm distances are not resolvable by widefield microscopy techniques and hence commenting on distance variations as small as a change from 216nm to 230nm is not reliable.

d. The violin plots of distances between loci have values below zero as well - is this possible?

3. dCas9 pull-down experiment: This is an excellent experiment for discovering proteins bound to these clusters. However, since this requires formaldehyde crosslinking, do the authors have confidence that any cluster-specific protein will still remain bound and be pulled-down? Authors need to clearly discuss this caveat of the experimental design.

4. Role of sigmaH in rrn operon clustering. Given that there could be global transcriptional changing during heat shock/ sigmaH overexpression, the decrease in rrn clustering could be due to more global changes to chromosome organization. Have the authors conducted experiments to see whether the sigmH-dependent declustering is specific to rrn operons, or whether such over-expression triggers more global changes in chromosome organization/ transcription (voa RNA-seq/ chromosome capture methods)?

5. Discussion section: This section is rather long, and mostly recapitulates the results. Authors should tighten this section significantly. The model suggests that sigmaH binds the inner-membrane and causes declustering. However, I am not clear as to how the inner membrane binding would cause this declustering. The model figure does not provide much insight into this proposed mechanism.

Minor comments:

1. Scale bar is missing in microscopy images

2. Please provide details of the number of biological replicates in each of the violin plots. Were these data pooled across replicates?

3. Upset plot in Fig. 3A is not readable

4. Please check the grammar for the discussion section L254-258.

Reviewer #2: The E.coli chromosome has seven ribosomal RNA operons and six of them show clustered spatial localization in the cell. In this manuscript by Ho and Harshey, the authors perturbed the cell using physical and chemical stresses including amino acid starvation, heatshock, coldshock and ethanol shock, and found that ethanol shock and heatshock disrupted the clustering. The authors went on the show that it is the elevated level of the sigma factor RpoH that disrupts the clustering of rrn operons. A similar effect was observed when FecI was expressed. Both RpoH and FecI associate with the cell membrane and both were counteracted by expression of sigma factor RpoD. The authors concluded rrn operons are clustered at the cell membrane directly or indirectly by RpoD.

This is a solid study involving a complementary approaches. The experiments were carefully executed, the results were critically analyzed and interpreted. The topic will be of broad interest to researchers that study chromosome biology field. However, I found some major gaps of understanding (see below). Filling these gaps could strengthen the authors' model:

1) One important component of the model is rrns are clustered at the cell membrane. The authors have a lot of images for the localization of rrns in the cells. When analyzing these data, is there evidence that the rrn loci are associated with the cell membrane? In other words, is there a tendency for the foci to be at the periphery of the cell?

2) Figure 2D. Why does heatshock reduce the copy number of rrnA but not rrnD? Because the number of foci are different, the absence of colocalization might be overestimated. In Figure 2CD or an SI figure, the authors should show a different pair of loci that have similar number of foci before and after RpoH expression.

3) GroEL is pulled down from 4/7 rrn operons. For dCas9 pulldown. It is unclear whether GroEL specifically get enriched at the rrn operons or it just has higher levels upon stress and get enriched everywhere. To distinguish these two possibilities, I suggest the authors do repeat dCas9 pulldown using one or two control loci that are not close to any rrn and examine GroEL enrichment.

4) Figure 6: Mu experiment. It looks like the strongest effect RpoH has is on the overall landscape of genome folding, and not specifically affecting rrn clustering. Then the question is: does RpoH specifically disrupt rrn flustering or does it lower chromosome interactions overall and the declustering of rrns is just a consequence of chromosome reorganization? Figure 6E-J showed lower interactions between rrnD and other rrns upon RpoH expression, does rrnD have lower interactions with non-rrn sites as well? If the authors have a Mu inserted in a non-rrn region (say geneX), then look at its interactions with regions (gene y, z, w, t etc) that are the same distance away as rrns away from rrnD, then measure the interaction between x-y, x-z, x-w, x-t, etc, before and after RpoH expression, does RpoH lower the interaction of these other loci pairs as well?

5) Figure 7 model: Where should be GroEL put in this model? RpoD not only bind to rrn promoters, but also bind to promoters of many housekeeping genes. What is preventing these other genes getting incorporated into the rrn clusters? Does the model predict that other highly transcribe genes (outside of rrn operons) are clustered with the rrns? Does Mu result support the idea the higher the expression of a gene, the more interactions it has with a rrn?

Reviewer #3: Review of Ho and Harshey, clustering of rRNA operons in E. coli is disrupted by sigma H

This manuscript is a follow up of the Harshey lab's seminal paper published in Cell in 2020, Walker et al. 180: 703-716. In that work, the authors found that 6 of the 7 rRNA operons in an E. coli cell (all except rrnC) are clustered in space in vivo. The authors find, using the Phage Mu method employed in the Cell paper and also by fluorescence microscopy using tags on the rRNA operons on the chromosome, that this clustering phenomenon is disrupted by heat shock or by expression of sigma 32 or sigma FecI, two inner membrane-associated sigma factors, but not by other sigma factors. Even a small increase in sigma 32 levels is sufficient for de-clustering, but transcription by Esigma32 is not required. Co-expression of sigma 70 with sigma 32 is sufficient to rescue clustering, suggesting that sigma 32 competes with sigma 70 to cause de-clustering of the physical contacts between the rRNA promoter regions. In an attempt to use mass-spec to identify proteins that cause clustering, the authors identify GroEL as making a contribution to clustering of 4 of the 6 clustered operons. The authors address other possible contributors, including condensate formation, concluding that RNAP condensates are a feature of highly active transcription units, but since neither NusB or RpoZ impact clustering whereas they are crucial for condensate formation, condensates are not responsible for rRNA operon clustering. In general, I found this paper very clear and well-written. I have only very very minor comments.

1. In many figures, fluorescent images are superimposed on phase contrast images. However, these are referred to as "bright field". My understanding is that these images are phase contrast, not "bright field" (which does not mean the same thing as phase contrast). Please correct the text, legends, and methods.

2. I had some difficulty determining on line 136 which panel in Fig. 3 the authors are referring to. I think it's the "red" plot that shows the increased distance, but describing this as the third plot is a bit confusing.

3. On line 139 the authors refer to Fig. 2D. Should this be Fig. 3D?

4. Would it be useful to point out in the text that the numbers above the plots in Figures 2B, 3D and E, and 4A and C are the numbers of pairs monitored, not the average distances between the rRNA operons? I found this confusing at first. An alternative solution would be to move the number of measured foci pairs to the legend or Methods (just saying 300-800 would be sufficient) and including the actual distances measured on the figure, so they match the text.

5. Line 357. The order of RNAP subunit assembly is alpha2 � alpha2- beta �alpha2- beta- beta prime � alpha2- beta- beta prime- omega. The earlier referenced Ishihama pathway was before recognition of the importance of omega for assembly and was superseded by the later reference (ref. 51). Including the earlier pathway might be confusing.

6. Fig 1. Perhaps it should be mentioned in the legend that the positions of the operons on the circular chromosome are not to scale.

7. The generic E. coli rRNA operon shown in Fig. 1 is pictured without a distal tRNA and is labeled as having a "variable tRNA" in the 16S-23S spacer. Three E. coli rRNA operons have a tRNA or two distal to the 23S or 5S genes. Perhaps to be more precise, it might be best to label the schematic operon as rrnG and indicate the identity of the spacer tRNA as tRNAGlu2 and also say that some operons have distal tRNAs.

8. I found a couple of typos: In ref. 37, Ciaran is the author's first name, the last name is Condon. Also, on line 400, the text should be "use of non-Mu" …

---

## [Decision Letter · Decision Letter 2]

21 Feb 2025

Dear Dr Harshey,

Thank you for your patience while we considered your revised manuscript "Clustering of rRNA operons in E. coli is disrupted by σH" for publication as a Research Article at PLOS Biology. This revised version of your manuscript has been evaluated by the PLOS Biology editors, the Academic Editor and two of the original reviewers.

Based on the reviews and on our Academic Editor's assessment of your revision, we are likely to accept this manuscript for publication, provided you satisfactorily address the remaining editorial points. Please also make sure to address the following data and other policy-related requests.

a) We routinely suggest changes to titles to ensure maximum accessibility for a broad, non-specialist readership, and to ensure they reflect the contents of the paper. Please ensure you change both the manuscript file and the online submission system, as they need to match for final acceptance:

"Membrane-associated σ factors disrupt rRNA operon clustering in E. coli"

b) We would like to encourage to change the article type to a Short Report. Short Reports are required to have 4 main figures, while you currently have 7. Could you please combine some of the current figures or send some to the Supplementary material?

Please supply the numerical values either in the a supplementary file or as a permanent DOI’d deposition for the following figures:

Figure 2B, 3BCD, 4A-D, 5ACD, 6A-J, S2, S3, S4, S5, S6ABC

d) Please ensure that you are using best practice for statistical reporting and data presentation. These are our guidelines https://journals.plos.org/plosbiology/s/best-practices-in-research-reporting#loc-statistical-reporting and a useful resource on data presentation https://journals.plos.org/plosbiology/article?id=10.1371/journal.pbio.1002128

- If you are reporting experiments where n ≤ 5, please plot each individual data point.

e) Please cite the location of the data clearly in all relevant main and supplementary Figure legends, e.g. “The data underlying this Figure can be found in S1 Data” or “The data underlying this Figure can be found in https://doi.org/10.5281/zenodo.XXXXX”

f) Please ensure that your Data Statement in the submission system accurately describes where your data can be found and is in final format, as it will be published as written there.

g) Per journal policy, if you have generated any custom code during the course of this investigation, please make it available without restrictions upon publication. Please ensure that the code is sufficiently well documented and reusable, and that your Data Statement in the Editorial Manager submission system accurately describes where your code can be found.

We expect to receive your revised manuscript within two weeks.

*Published Peer Review History*

*Press*

Sincerely,

Melissa

Melissa Vázquez Hernández, PhD

Associate Editor

PLOS Biology

on behalf of

Richard

Richard Hodge, PhD

Senior Editor

rhodge@plos.org

PLOS Biology

REVIEWERS' COMMENTS:

Reviewer #1: Authors have addressed all my comments.

Reviewer #2: The authors have done a great job in the revision. The added analyses and edits greatly improved the manuscript. I have no further concerns.

---

## [Editor Report · Decision Letter 3]

13 Mar 2025

Dear Dr Harshey,

Thank you for the submission of your revised Short Reports "Membrane-associated σ factors disrupt rRNA operon clustering in E. coli" for publication in PLOS Biology. On behalf of my colleagues and the Academic Editor, Michael T. Laub, I am pleased to say that we can in principle accept your manuscript for publication, provided you address any remaining formatting and reporting issues. These will be detailed in an email you should receive within 2-3 business days from our colleagues in the journal operations team; no action is required from you until then. Please note that we will not be able to formally accept your manuscript and schedule it for publication until you have completed any requested changes.

IMPORTANT: Many thanks for citing the location of the raw data in the legend of Figure 1. However, we require this is done for every figure where raw data is available saying “The data underlying this Figure can be found in Supplemental “Raw Data” file”. I have asked my colleagues to include this request alongside their own.

PRESS

Sincerely, 

Melissa

Melissa Vázquez Hernández, PhD

Associate Editor

PLOS Biology

on behalf of

Richard Hodge, PhD, PhD

Senior Editor

PLOS Biology

rhodge@plos.org